# Non-Inflamed Tumor Microenvironment and Methylation/Downregulation of Antigen-Presenting Machineries in Cholangiocarcinoma

**DOI:** 10.3390/cancers15082379

**Published:** 2023-04-20

**Authors:** Naoshi Nishida, Tomoko Aoki, Masahiro Morita, Hirokazu Chishina, Masahiro Takita, Hiroshi Ida, Satoru Hagiwara, Yasunori Minami, Kazuomi Ueshima, Masatoshi Kudo

**Affiliations:** Department of Gastroenterology and Hepatology, Faculty of Medicine, Kindai University, 377-2 Ohno-higashi, Osaka-sayama 589-8511, Osaka, Japan

**Keywords:** cholangiocarcinoma, tumor microenvironment, methylation, downregulation, antigen-presenting machinery, human leukocyte antigen, driver mutation

## Abstract

**Simple Summary:**

Cholangiocarcinoma (CCA) is a chemotherapy-resistant cancer. Reportedly, combination chemotherapy that includes immune checkpoint inhibitors (ICIs) showed superior effects compared to conventional chemotherapy for the treatment of advanced CCA. Although the efficacy of ICIs is considered to be affected by the tumor immune microenvironment (TME), the genetic/epigenetic background associated with TME in CCA is unknown. Here, we find that downregulations of genes involved in antigen presentation are associated with a “non-inflamed” tumor; the downregulations are inversely correlated with their DNA methylation levels. All tumors in the “inflamed” group exhibited an upregulation/low-methylation pattern. In contrast, the majority of CCAs in the non-inflamed group represent a downregulation/high-methylation pattern in antigen-presenting machineries. In addition, unique gene mutations of CCA that should induce DNA methylation events, like those in IDH1/2, are exclusively observed in CCAs carrying a non-inflamed tumor phenotype. These results may prove to be a clue to develop novel biomarkers and combination therapies for the treatment of CCA using ICIs.

**Abstract:**

Cholangiocarcinoma (CCA) is a refractory cancer; a majority of CCAs represents a non-inflamed tumor phenotype that should be resistant to treatment, including immune checkpoint inhibitors (ICIs). In this study, we aimed to understand the molecular characteristics associated with non-inflamed CCAs. The genetic/epigenetic status of 36 CCAs was obtained from the Cancer Genome Atlas (PanCancerAtlas). CCAs were classified based on immune class using hierarchical clustering analysis of gene expressions related to tumor-infiltrating lymphocytes. The associations between immune class and genetic/epigenetic events were analyzed. We found that the tumors with alterations in FGFR2 and IDH1/2 had a “non-inflamed” tumor phenotype. A significant association was observed between the non-inflamed group and the downregulation of genes involved in antigen presentation (*p* = 0.0015). The expression of antigen-presenting machineries was inversely correlated with their DNA methylation levels, where 33.3% of tumors had an upregulation/low-methylation pattern, and 66.7% of tumors had a downregulation/high-methylation pattern. All tumors in the “inflamed” group exhibited an upregulation/low-methylation pattern. In contrast, 24 of 30 tumors in the non-inflamed group represent the downregulation/high-methylation pattern (*p* = 0.0005). Methylation with downregulation of antigen-presenting machineries is associated with the “non-inflamed” tumor phenotype of CCAs. This evidence provides important insights for developing new strategies for treating CCA.

## 1. Introduction

Cholangiocarcinoma (CCA) is a malignancy that develops from the intrahepatic and extrahepatic bile ducts and is categorized as intrahepatic cholangiocarcinoma (IHC), perihilar cholangiocarcinoma (PHC), and distal cholangiocarcinoma (DC) [1,2]. It is known that inflammation of the bile duct causes the emergence of CCA. For example, bile duct cystic disorders, pancreaticobiliary maljunction, biliary–enteric drainage, hepatobiliary flukes, and hepatolithiasis, all induce recurrent cholangitis. These are the well-known risks of CCA. In addition, an association between primary sclerosing cholangitis and CCA, especially PHC, has been reported. More importantly, risk factors of hepatocellular carcinoma (HCC), such as liver cirrhosis, chronic hepatitis B and hepatitis C, and diabetes, are also risks of IHC [2,3].

Although surgical resection is the only curative treatment for CCA, most patients are diagnosed at the advanced stage of the disease, in which curative surgery is not suitable due to nonspecific symptoms in the early stage of the disease [2]. Therefore, effective systemic chemotherapy is crucial for improving the prognosis in CCA cases. The combination of gemcitabine and cisplatin (GC) has been used as a standard chemotherapy for advanced CCA [4]. A recent phase III trial also showed the superior efficacy of GC treatment in combination with tegafur, gimeracil, and oteracil potassium, commonly referred to as TS-1, over GC treatment in terms of the survival of patients with unresectable or recurrent CCA [5].

In addition to conventional anticancer drugs, molecular-targeted agents (MTAs) targeting the driver mutations responsible for this type of tumor showed promising efficacy for advanced CCA with disease progression after first-line chemotherapy [6]. For example, based on phase II/III clinical trials, anti-fibroblast growth factor receptor (FGFR) tyrosine kinase inhibitors and isocitrate dehydrogenase (IDH) inhibitors have been reported to be effective in tumors with *FGFR2* gene fusions or rearrangements and in tumors with *IDH1* mutations, respectively [7,8,9,10]. In addition, a combination of immune checkpoint inhibitors (ICIs), anti-programmed cell death ligand-1 antibodies, and GC treatment has been reported to significantly improve tumor response and survival compared with GC therapy alone in patients with previously untreated, unresectable, or metastatic CCA [11].

Since MTAs and ICIs can be used to treat CCAs, it is essential to understand the tumor immune microenvironment (TME) and its association with driver mutations. The status of the driver mutation can be critical for treatments with MTAs, and the TME affects the tumor’s response to ICIs [12]. Reportedly, around 36% of IHCs can be classified in the inflamed class of the TME [13], whereas the majority of CCAs are considered as a non-inflamed tumor, for which immune-based therapies are unlikely to be effective [14]. Reports regarding the association between TME and driver mutation status indicate that activating mutations in Wnt/β-catenin are responsible, partly, for the establishment of a non-inflamed tumor phenotype in melanoma, colon cancer, and HCC, which has been attributed to the downregulation of C-C chemokine motif ligands (CCL) 4 and 5 [15,16]. However, mutation in Wnt/β-catenin pathway genes is uncommon in CCA [17], suggesting that other genetic/epigenetic changes may contribute to the establishment of non-inflamed tumor phenotype [18]. In this study, we focused on this critical issue and analyzed the events that were associated with the non-inflamed class in CCA, which should be critical information for the effective application of ICIs for advanced CCAs.

## 2. Materials and Methods

### 2.1. Patient Data

Genomic data from 1576 cholangiocarcinoma samples from eleven cohorts were deposited in the cBioPortal public database (https://www.cbioportal.org, accessed on 3 November 2022). All of the following data, the status of gene mutations, structural variants, putative copy number alterations by whole-exome sequencing, mRNA expression levels by RNA sequencing, and DNA methylation levels by β-values from Infinium^®^ Human Methylation (HM) 450 BeadChip were available for 36 tumors from 36 patients used in this study (TCGA PanCancer Atlas) (Appendix A). In this cohort, 29 patients had IHC and four had PHC. Among the remaining three, two were shown as extrahepatic cholangiocarcinoma and one was listed as cholangiocarcinoma, respectively. Sixteen patients were male, and twenty were female. Thirty-one were white, three were Asian, and two were black or African-American. Regarding distant metastasis or spread, according to the American Joint Committee (AJCC) on Cancer Metastasis Stage Code, 28 were M0, 5 were M1, and 3 were MX. For the tumor stage, according to the AJCC, 19 were T1, 6 were T2, 5 were T3, 4 were T2B, and 2 were T2B. Patients and the public were not involved in this study’s design, conduct, reporting, or dissemination.

### 2.2. Statistical Analysis

Fisher’s exact test was used to compare the categorical variables. For comparing two contentious variables, the non-parametric Wilcoxon rank-sum test was applied. For comparisons of multiple contentious variables of mRNA expressions and DNA methylation levels among the three TME categories, we applied the non-parametric Steel–Dwass test. Hierarchical clustering analysis was performed to identify a specific CCA cluster based on transcription and methylation levels. *Z*-scores were applied for mRNA expression and DNA methylation levels for normalization among multiple genes. The correlation between the methylation level and the expression of the corresponding gene was evaluated using Pearson’s correlation coefficient. All *p-*values were two-sided, and a *p* < 0.05 was considered statistically significant. All statistical analyses were performed using JMP software version 16.2.0 (SAS Institute Inc., Cary, NC, USA).

## 3. Results

### 3.1. Classification of CCAs Based on Transcriptome Data

A hierarchical clustering analysis using mRNA levels in tumor-infiltrated lymphocytes (TILs) was applied to classify the CCAs according to the TME type. The following genes were selected based on a previous study [19]: *CD2*, *CD3D*, *CD8A*, *CD8B*, *CD48*, *CD52*, and *CD53* as T-cell surface markers; *FYB1*, *IFNG*, *LAPTM5*, *LCP2*, *PTPRC*, and *SLA* as T-cell signaling and activation-related markers; *CCL4*, *CCL5*, *CXCL9*, *CXCL10*, *CXCL11*, and *CXCR4* as lymphocyte chemotactic markers; *GZMA*, *GZMB*, *GZMK*, *GZMH*, and *GZMM* as T-cell-related cytolytic factors; *CTLA4*, *LAG3*, *TIGIT*, *CD274*, *PDCD1*, and *HAVCR4* as immune checkpoint molecules (Figure 1). The hierarchical clustering analysis with 36 CCAs formed three major clusters; 6 (17%) formed a cluster with prominent expression of these TIL-related markers and were defined as the “inflamed” tumor group because tumors in this cluster were considered as carrying the highest degree of TILs compared to other groups (Appendix A). We classified the remaining 30 tumors as the “non-inflamed” group. Within this group, 17 tumors exhibited a cluster with low expression of TIL-related markers and were defined as “non-inflamed A”, and 13 tumors formed a cluster characterized with a mild expression of these markers and were defined as “non-inflamed B”. The distributions of *z*-scores of mRNA expression of T-cell surface markers, T-cell signaling and activation-related markers, lymphocyte chemotactic markers, T-cell-related cytolytic factors, and immune checkpoint molecules are shown in Appendix A.

### 3.2. Genetic and Epigenetic Alterations Associated with the Inflammation Profile of CCAs

The association between the genetic and epigenetic alterations frequently observed in CCAs was then analyzed (Table 1). Regarding the driver mutations, we identified alterations in the *FGFR2* gene in seven cases (7/36, 19.4%), fusion genes in five cases and mutations in two cases. Similarly, eight mutations are detected in the *BAP1* genes (8/36, 22.2%), seven in the *IDH1*/*2* genes (7/36, 19.4%), six in the *ARID1A*/*2* genes (6/36, 16.7%), and five in the *TP53* genes (5/36, 13.9%). In addition, several mutations are detected in the genes involved in antigen presentation, including one in *HLA-A* and five in *HLA-B* (6/36, 16.7%). Four cases show copy number loss at the HLA locus (6p21.3). The median value of tumor mutation burden (TMB) score, which is determined by the number of nonsynonymous mutations/megabase, is 1.45 (range, 0.2–21.8). We defined a TMB score of >2.6 as “high TMB” and a TMB score of <2.6 as “low TMB”, which corresponded to the 75th percentile of cases. Nine cases have tumors with high TMB and 27 with low TMB. We also evaluated the expression and methylation levels of the genes involved in antigen presentation, including *HLA-B*, *HLA-C*, *HLA-E*, *B2M*, *TAP-1*, and *CIITA*, which were previously described in HCC cases [19]. We performed hierarchical clustering analysis to classify tumors based on their gene expressions. Of the 36 CCAs, 14 (38.9%) are classified into the “upregulation” and 22 (61.1%) into the “downregulation” groups for the expression of antigen-presenting machineries, respectively (Appendix A).

To find the genetic and epigenetic backgrounds involved in the TME in CCAs, we analyzed the association between inflammation class determined with hierarchical clustering analysis (shown in Figure 1) and genetic/epigenetic alterations (Table 1). Although statistically insignificant, we found that all seven cases with alterations in the *FGFR2* gene and all seven cases with mutations in the *IDH1*/*2* gene are classified into the “non-inflamed” group. There is no association between TMB score and the inflammation class of tumors. Intriguingly, a significant association between downregulation of antigen-presenting machineries and the non-inflamed tumor phenotype is observed; all 22 cases with downregulation of these genes are classified into the “non-inflamed” group (*p* = 0.0015, Table 1).

### 3.3. The Immunological Microenvironment of “Non-Inflamed” Is Associated with Methylation and Downregulation of Antigen-Presenting Machineries and Mutations in IDH Genes in Biliary Tract Cancer

Since 22 of 36 (61.1%) CCAs were in the “downregulation” groups of genes involved in the antigen presentation (Appendix A), we analyzed the association between the downregulation and mutations of the *HLA-A*, *-B*, and *-C* genes. In addition, we examined copy number loss on chromosome 6p21.3, where the HLA genes are located. We found no association between the downregulation of the antigen-presenting machineries and mutations and copy number losses of HLA. Next, to determine whether epigenetic events were associated with the downregulation of the antigen-presenting machineries, we compared gene expression and methylation levels and found that these levels were inversely correlated with each other (*p* = 0.0005, *r*^2^ = 0.3036 for *HLA-B*; *p* < 0.0001, *r*^2^ = 0.3690 for *HLA-C*; *p* < 0.0001, *r*^2^ = 0.4693 for *HLA-E*; *p* = 0.0393, *r*^2^ = 0.11906 for *B2M*; *p* = 0.0001, *r*^2^ = 0.3565 for *TAP1*; *p* = 0.0291, *r*^2^ = 0.1325 for the *HLA-B*, Figure 2). We further classified the tumors into two clusters based on gene expression and methylation levels of the antigen-presenting machineries using hierarchical clustering analysis (Figure 3). Thus, 12 cases show an upregulation/low-methylation pattern (12/36, 33.3%) and 24 cases represent a downregulation/high-methylation pattern (24/36, 66.7%). All the six tumors with an inflamed phenotype show the upregulation/low-methylation pattern in the antigen-presenting machineries. In contrast, 24 of 30 tumors (80%) that show a non-inflamed phenotype represent the downregulation/high-methylation pattern (*p* = 0.0005, Figure 3). In addition, expression levels are significantly lower in the non-inflamed group than in the inflamed group for all genes involved in the antigen presentation (*p* = 0.0096, 0.0416, 0.0006, 0.0039, 0.0026, and 0.0004 for the *HLA-B*, *HLA-C*, *HLA-E*, *B2M*, *TAP1*, and *CIITA*, respectively, Figure 4a). Similarly, methylation levels are significantly higher in the non-inflamed group than in the inflamed group for all the genes (*p* = 0.0030, 0.0081, 0.0109, 0.0066, 0.0075, and 0.0175 for the *HLA-B*, *HLA-C*, *HLA-E*, *B2M*, *TAP1*, and *CIITA*, respectively, Figure 4b). Because we further divided the non-inflamed tumors into non-inflamed A and non-inflamed B, in which mRNA expressions of TIL-related markers were lower in the non-inflamed A group than in non-inflamed B, we compared mRNA expressions and DNA methylation levels among the three groups. The inflamed group shows the highest mRNA expression and the lowest DNA methylation level (Appendix A). In addition, median mRNA expression levels are lower in tumors in non-inflamed A than those in non-inflamed B for all six antigen-presenting machineries but TAP1 (Appendix A). Similarly, median DNA methylation levels are higher in tumors in non-inflamed A than those in non-inflamed B in all six antigen-presenting machineries but B2M and CIITA (Appendix A). We also examined the distribution of *IDH* mutations and *FGFR2* alterations for the expression and methylation status in antigen-presenting machineries. All the seven tumors carrying mutations in *IDH* genes and six of the seven tumors carrying *FGFR2* alterations show the downregulation/high-methylation pattern (Figure 3).

## 4. Discussion

So far, several clinical trials have been conducted for the treatment of CCA [2]. Importantly, clinical trial studies reveal that two kinds of MTA, the FGFR tyrosine kinase inhibitor and the IDH1 inhibitor, were reported to be effective in treating advanced chemotherapy-resistant CCAs after progression on first-line systemic therapy with *FGFR2* fusions/rearrangements and *IDH1* mutations, respectively [7,8,9,10]. In addition, HER2 and BRAF inhibitors are undergoing incorporation into the conventional systemic chemotherapy [20].

Recently, treatment with ICIs (durvalumab) in combination with gemcitabine and cisplatin showed superior efficacy to conventional first-line treatment with gemcitabine and cisplatin in overall survival and progression-free survival in patients with advanced CCAs [11]. In addition, two kinds of MTA, the FGFR tyrosine kinase inhibitor and the IDH1 inhibitor, were reported to be effective in treating advanced chemotherapy-resistant CCAs with *FGFR2* fusions/rearrangements and *IDH1* mutations, respectively [7,8,9,10]. The efficacy of ICIs should be affected by the TME, and the efficacy of MTAs depends on the mutation status of the target genes; understanding the TME and its association with driver mutations is necessary for chemotherapy management [21,22,23].

Several studies have demonstrated an immunosuppressive TME and activation of specific oncogenic signaling pathways in malignant tumors and reported potential treatment strategies for combinations of ICIs and MTAs [14,24,25]. For example, activating mutations in *KRAS*, *BRAF*, and *PI3KA*, commonly detected in many cancers, may play a role in immune escape through the upregulation of programmed cell death ligand-1 and immunosuppressive cytokines [21,26]. The activation of the Wnt/β-catenin pathway by *CTNNB1* mutations contributes to a decline in TIL by inducing downregulation of the chemokines CCL4 and CCL5 that attract dendritic cells and T cells into tumor tissue [15,16]. Thorsson et al. performed an integrated analysis of the TME with more than 10,000 tumors from 33 cancer types, and found that driver mutations, such as those found in *CTNNB1* and *IDH1*, were linked to the decreased number of TILs [24].

Regarding HCC cases, Montironi et al. characterized the immunogenomic context of tumors and classified 37% of the tumors into an “inflamed” group and 63% into a “non-inflamed” group, where a majority of *CTNNB1* mutations were associated with non-inflamed tumors [19]. In a mouse HCC model, mutations in the Wnt/β-catenin pathway were shown to cause a decrease in CCL5 and the exclusion of TILs. The activation of this pathway has been reported to be associated with poor response to treatment with ICIs in HCC [16,27,28,29]. We previously reported that Wnt/β-catenin activation, TIL degree, and programmed cell death ligand-1 expression could predict the anti-tumor response of ICI monotherapies in patients with HCC [29]. This suggests that the genomic status is an important contributor to the establishment of cancer-specific TMEs, and the immunogenomic background should be considered for drug therapy using ICIs [25,26]. Because intrahepatic bile ducts exist in Glisson’s capsules that is affected by chronic hepatitis, inflammation of this area may well cause the proliferation of the progenitor cells that lead to the emergence of IHC as well as HCC. From this point of view, there should be a molecular spectrum between them. Jeon et al. reported four liver cancer subtypes from RNA-sequencing profiles, typical HCC, IHC-like HCC, HCC-like IHC, and IHC types, that are further subdivided into small duct and large duct type [30]. They showed transcriptomic, genomic, and radiopathologic features unique to each subtype. Interestingly, the HCC-like IHC subtype is associated with HCC-related etiologic factors.

In contrast, the profile of driver mutations in CCAs differs from that in HCCs. *FGFR2* fusions/rearrangements and mutations in *IDH1*/*2*, *BAP1*, and *KRAS* are frequently detected in CCAs [17,18,31]. Recent studies showed that the mutations commonly observed in CCAs are associated with the type of TME [32]. Mody et al. performed an integrative cluster analysis of CCAs and showed that *FGFR2* fusions were inversely associated with tumor immune infiltration [22]. Martin-Serrano et al. classified IHCs based on the TME type and showed that 63% of the cases were classified into a “non-inflamed” phenotype; *FGFR2* fusions were mainly observed in the “non-inflamed” group [13]. Lin et al. also reported that *FGFR2* mutations were correlated with lower immune infiltration [33]. From this point of view, FGFR tyrosine kinase inhibitors should be more effective than ICIs if the tumors carry *FGFR2* alterations because tumors with alterations in the *FGFR2* genes generally showed a “non-inflamed” tumor phenotype [8]. In contrast, the association between mutations in *IDH1*/*2* and TME is still controversial. Studies reported that *IDH1*/*2* mutations could shape a “cold immune microenvironment” [34,35]. Martin-Serrano et al. showed that CCAs with *IDH1*/*2* mutations were mainly characterized by the “non-inflamed” tumor phenotype with a hepatic stem-cell-like future [13]. On the contrary, Lin et al. reported that tumors with *IDH1*/*2* mutations showed T-cell infiltration in HBV-related CCAs [33].

In this study, we found that “non-inflamed” CCA is strongly associated with the downregulation of genes related to the antigen-presenting machineries, which consists mainly of HLA molecules (Figure 5). In addition, the majority of antigen-presenting machineries showed the lowest mRNA expressions and the highest DNA methylation levels in the tumors in the non-inflamed A group where downregulation of TIL-related markers was the most prominent (Appendix A). Previous reports show that alterations related to the antigen-presenting machineries could induce an immune-cold phenotype in various cancers. Somatic alterations in HLA-class I molecules are thought to cause immune escape from cytotoxic T lymphocytes in colorectal cancer [36]. An association between the allele-specific loss of HLA and immune escape has also been reported in lung cancer [37]. Based on the study of the immunologic subclassification of colorectal cancers with high microsatellite instability, the decrease in TILs is partly attributed to the downregulation of *HLA-A*, *HLA-B*, and *HLA-C*. In contrast, methylation of antigen-presenting machineries is inversely associated with gene expression in breast cancer [38]. In this study, we also focused on the alterations in the antigen-presenting machineries in the context of TME in CCAs. Although we did not find any association between the TME and mutations in *HLA-*related genes or copy number losses in HLA loci, which are frequently detected in CCAs (Table 1 and Appendix A), the expressions of the antigen-presenting machineries are significantly associated with the “non-inflamed” phenotype of CCAs, which are inversely correlated with their DNA methylation levels (Figure 2). Importantly, clustering analyses based on gene expression and methylation levels of these genes revealed that 24 of the 30 CCAs in the “non-inflamed” group showed a downregulation/high-methylation pattern, and all 6 CCAs in the “inflamed” group showed an upregulation/low-methylation pattern. This evidence suggests that methylation-induced downregulation of the antigen-presenting machineries may be an important factor in the establishment of a “non-inflamed” tumor phenotype in CCAs.

We also analyzed the association between driver mutations recurrently detected in CCAs and the expression/methylation patterns of antigen-presenting machineries. Interestingly, all seven cases with tumors carrying mutations in *IDH1*/*2* show the downregulation/high-methylation pattern. It is known that IDH 1 and 2 are essential enzymes in converting isocitrate to α-ketoglutarate (α-KG). Mutations in *IDH1* and *IDH2* have been reported to cause DNA hypermethylation because mutant IDH induces the conversion of α-KG to D-2-hydroxyglutarate, which is a potent inhibitor of α-KG-dependent DNA demethylases and ten–eleven translocation enzymes that catalyze the iterative demethylation of 5-methylcytosine [39,40]. Therefore, mutations in *IDH1*/*2* may induce DNA hypermethylation in the antigen-presenting machineries and contribute to the establishment of the “non-inflamed” CCAs. Currently, IDH inhibitors and ICI-based therapies are approved for the treatment of CCAs. Therefore, it is attractive to speculate that IDH inhibitors may restore DNA demethylation and expression of the antigen-presenting machineries, which can improve the efficacy of ICIs.

We also found no tumors with *FGFR2* mutations in the “inflamed” group. Interestingly, Jasakul et al. performed integrative clustering of CCAs and found that clusters enriched in IDH1/2 mutations and FGFR alterations were characterized by hypermethylation at the CpG shore [18]. Previous reports showed that activating the FGF signaling could inhibit the IFN-γ signaling pathway with activation of the suppressor of cytokine signaling 1 (SOCS1) [41] and induces downregulation of β2-microglobulin (B2M) through several mechanisms including DNA methylation [42]. Therefore, it may be possible that FGFR2 alterations might also contribute to the establishment of a “non-inflamed” tumor phenotype through the downregulation of antigen-presenting machineries. Because both mutations in the *IDH1*/*2* and alterations in *FGFR2* genes may contribute to the suppression of genes involved in antigen-presenting machineries, we analyzed the association between the mutations/alterations in either of these 2 genes and expression status of antigen-presenting machineries, and found a significant association, where 11 of 12 CCAs with the mutations/alterations in either of these 2 genes are members of the downregulation group, and only 1 of 12 with such mutations/alterations is in the upregulation group (*p* = 0.0111, Appendix A).

Our study has some limitations, despite intensive analyses of mutations, gene expression, and methylation events in CCAs and their association with “non-inflamed” tumor phenotypes. First, because of the small number of samples in the cohort of CCAs studied, we could not find a significant association in some analyses, including the association between mutations in *IDH1*/*2* and the TME. Second, although the mutation profile differed at the different tumor sites of origin, we could not perform a stratified analysis between IHC, PHC, and DC [43,44]. Third, there are no clinical data on the efficacy of ICIs and FGFR and IDH1 inhibitors in “inflamed” and “non-inflamed” CCAs. Nevertheless, the results shown here suggest that the expression and methylation levels of antigen-presenting machineries are crucial for determining the type of TME in CCAs. Because CCA is a refractory malignancy with a high mortality rate, the results of this study should provide important insights in developing new strategies for the treatment of advanced CCAs.

## 5. Conclusions

We found that DNA methylation and downregulations of genes involved in antigen presentation are associated with a “non-inflamed” tumor phenotype. In addition, unique gene mutations of CCA that should induce DNA methylation events, such as those in *IDH1*/*2*, are exclusively observed in tumors showing a non-inflamed tumor phenotype. These results may prove a clue to develop novel biomarkers and combination therapies for the treatment of CCA using ICIs.

## Figures and Tables

**Figure 1 cancers-15-02379-f001:**
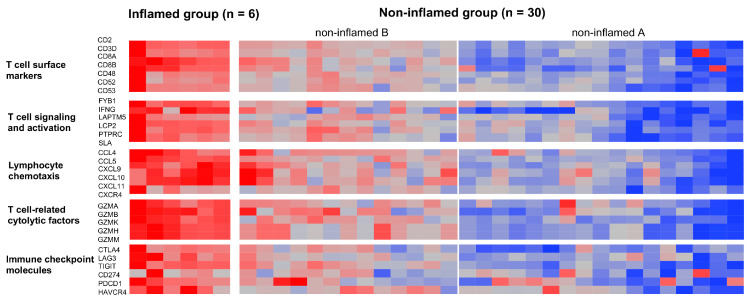
Hierarchical clustering analysis using z-scores of mRNA levels related to tumor-infiltrated lymphocytes. The color map (*red-gray-blue*): *red*, degree of increased mRNA expression; *blue*, degree of decreased mRNA expression. For clustering analysis, z-scores of mRNA levels of the following genes were used: CD2, CD3D, CD8A, CD8B, CD48, CD52, and CD53 as T-cell surface markers; FYB1, IFNG, LAPTM5, LCP2, PTPRC, and SLA as T-cell signaling and activation-related markers; CCL5, CXCL9, CXCL10, CXCL11, and CXCR4 as lymphocyte chemotactic markers; GZMA, GZMB, GZMK, GZMH, and GZMM as T-cell-related cytolytic factors; CTLA4, LAG3, TIGIT, CD274, PDCD1, and HAVCR4 as immune checkpoint molecules. Among the 36 cases, 6 (17%) formed a cluster and were considered to belong to the inflamed class because of the high expression of genes related to tumor-infiltrated lymphocytes. The remaining 30 cases were classified as non-inflamed. Of these, seventeen formed a cluster with low expression of these markers and were defined as non-inflamed A; thirteen cases with a mild expression of markers were defined as non-inflamed B.

**Figure 2 cancers-15-02379-f002:**
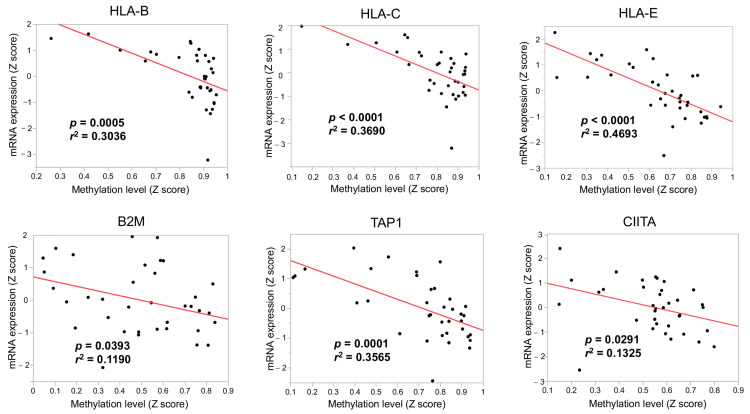
Correlation between methylation level and expression of the genes involved in antigen-presenting machineries. Negative correlations are observed between expression and methylation levels in HLA-B, HLA-C, HLA-E, B2M, TAP1, and CIITA. Pearson’s correlation coefficients are shown for each correlation.

**Figure 3 cancers-15-02379-f003:**
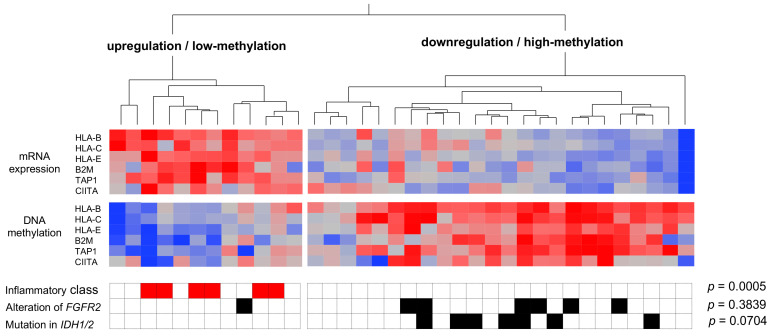
Hierarchical clustering analysis using z-scores of mRNA expression and methylation levels of antigen-presenting machineries. The color map (*red-gray-blue*): *red*, degree of increased mRNA expression or methylation density; *blue*, degree of decreased mRNA expression or methylation density. CCAs in the cluster with high expression (red rectangle) and low methylation levels (blue rectangle) were considered upregulation/low-methylation pattern, and those with low expression (blue rectangle) and high methylation levels (red rectangle) were determined as downregulation/high-methylation pattern, respectively. Tumors classified as inflamed group, which are indicated in Figure 1, are shown with red squares. Similarly, tumors harboring FGFR2 alterations and mutations in IDH1/2 are shown as black squares. All six tumors considered to have an inflamed tumor phenotype showed upregulation and low-methylation pattern in the genes involved in antigen presentation, whereas 24 of 30 tumors with a non-inflamed tumor phenotype showed downregulation with high-methylation pattern (*p* = 0.0005). All seven tumors carrying mutations in the IDH genes and six of the seven tumors carrying FGFR2 alterations were classified into the downregulation/high-methylation pattern group (*p* = 0.0704 and 0.3839 for mutations in the IDH genes and alterations in the FGFR2 genes, respectively).

**Figure 4 cancers-15-02379-f004:**
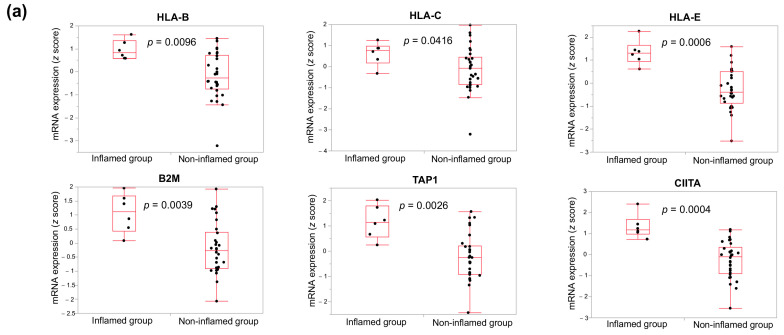
Comparisons of mRNA expression and DNA methylation level between inflamed and non-inflamed cholangiocarcinoma. mRNA expressions of the genes involved in antigen presentation are significantly higher in inflamed group than in non-inflamed group for all comparisons (**a**). Similarly, their DNA methylation levels are significantly lower in inflamed group than in non-inflamed group for all comparisons (**b**). Red boxes and whisker plots denote 75% and 95% distribution, respectively, and the red lines in the boxes show the median values. *p* values by non-parametric Wilcoxon rank-sum test are shown.

**Figure 5 cancers-15-02379-f005:**
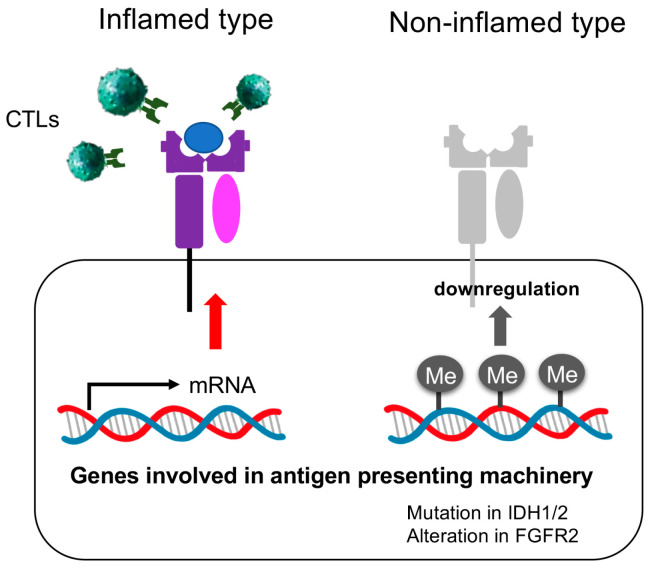
Schematic representation of the role of methylation and downregulation of genes involved in antigen presentation. In cholangiocarcinomas (CCAs), the phenotype of “inflamed” and “non-inflamed” classes, determined by the transcriptome of tumor-infiltrated lymphocytes, is strongly associated with the methylation and downregulation of genes related to antigen-presenting machinery. Consequently, lack of antigen-presenting machinery for the presentation of tumor-specific neoantigen result in the failure of attracting cytotoxic T lymphocytes (CTLs) into tumor. Some driver mutations unique to CCAs, such as mutations in the *IDH1*/*2* genes and alterations in the *FGFR2* gene, may be associated with these methylation and downregulation events.

**Table 1 cancers-15-02379-t001:** Association between immune class and genetic and epigenetic alterations in cholangiocarcinoma.

	Inflamed (*n* = 6)	Non-Inflamed (*n* = 30)	*p* Value by Fisher’s Exact Test
FGF2R alterations			
with (n = 7)	0	7	0.3171
without (n = 29)	6	23	
IDH1/2 mutations			
with (n = 7)	0	7	0.3171
without (n = 29)	6	23	
KRAS mutations			
with (n = 2)	0	2	1.0
without (n = 24)	6	28	
BAP1 mutations			
with (n = 8)	1	7	1.0
without (n = 28)	5	23	
TP53 mutations			
with (n = 5)	0	5	0.5638
without (n = 31)	6	25	
ARID1/2 mutations			
with (n = 6)	1	5	1.0
without (n = 30)	5	25	
HLA mutation			
with (n = 6)	1	5	1.0
without (n = 30)	5	25	
HLA copy number loss			
with (n = 4)	0	4	1.0
without (n = 32)	6	26	
TMB score ^1^			
low (n = 27)	5	22	1.0
high (n = 9)	1	8	
Expression of antigen-presenting machineries ^2^			
low (n = 22)	0	22	0.0015
high (n = 14)	6	8	

^1^ TMB, tumor mutation burden. TMB score is determined by the number of nonsynonymous mutations/megabase. We defined a TMB score of >2.6 as “high TMB” and a TMB score of <2.6 as “low TMB”, which corresponds to the 75th percentile of cases. ^2^ Expression of antigen-presenting machineries is determined using hierarchical clustering analysis of gene expression related to antigen-presenting machinery, including *HLA-B*, *HLA-C*, *HLA-E*, *B2M*, *TAP-1*, and *CIITA* (Appendix A).

## Data Availability

All data used in this study are available in a public data base (TCGA PanCancerAtlus) at: https://www.cbioportal.org/study/summary?id=chol_tcga_pan_can_atlas_2018. The data were accessed on 3 November 2022.

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
