# Peer review of "Non-Inflamed Tumor Microenvironment and Methylation/Downregulation of Antigen-Presenting Machineries in Cholangiocarcinoma"

_cancers, 2023, doi:10.3390/cancers15082379_

Round 1

Reviewer 1 Report

Dear Author,

Overall the research article is described well.

Please follow the comments  from the editor.

Thanks.

Author Response

To Reviewer 1

[Reviewer's overall comment]
The research article title “Non-inflamed tumor microenvironment and methylation/downregulation of antigen-presenting machineries in cholangiocarcinoma” by Naoshi Nishida et al., described well. The author demonstrated that non-inflamed epigenetic changes provide a new method for treating cholangiocarcinoma (CCA). The author used a public database for the genetic and epigenetic modifications associated with the inflammatory profile of CCA.
Overall the research article needs minor changes before accepting the publication.

[Response]

Thank you for the positive comments for our manuscript.

We have learned reviewer's comments carefully and revised the manuscript as listed below.

[Inquiries 1]
1. Introduction can be elaborated about the risk factors and subtypes of CAA.

[Response 1]

Thank you for the kind suggestion. We agree the association between risk factors and subtypes of CAA is quite informative. We stated about this issue in introduction, page 2, line 50-57, as follow;

" It is known that inflammation of the bile duct causes an emergence of CCA. For example, bile duct cystic disorders, pancreaticobiliary maljunction, biliary-enteric drainage, hepatobiliary flukes, and hepatolithiasis, all of which induce recurrent cholangitis, are the well-known risks of CCA. In addition, association between primary sclerosing cholangitis and CCA, especially PHC, has been reported. More importantly, risk factors of hepatocellular carcinoma (HCC), such as liver cirrhosis, chronic hepatitis B and hepatitis C, and diabetes, are also risks of IHC."

We added a new reference for this topic in ref. 3;

Razumilava, N.; Gores, G.J. Cholangiocarcinoma. Lancet. 2014, 383, 2168-2179.

[Inquiries 2]
2. Author can explain the correlation studies between intrahepatic carcinoma and cholangiocarcinoma.

[Response 2]

We agree the discussion about the correlation studies between intrahepatic carcinoma and cholangiocarcinoma is important for understanding the pathogenesis of these two types of liver cancer.

We discussed this topic in discussion, page 9, line 310-317, as follow;

"Because intrahepatic bile duct exists in the Glisson’s capsules that is affected by chronic hepatitis, inflammation of this area may well cause the proliferation of the progenitor cells that lead to the emergence of IHC as well as HCC. From this point of view, there should be conferring a molecular spectrum between them. Jeon et al., reported four liver cancer subtypes from RNA-sequencing profiles, typical HCC, IHC-like HCC, HCC-like IHC, and IHC types that are further subdivided into small duct and large duct type.[30] They showed transcriptomic, genomic, and radiopathologic features unique to each subtype. Interestingly, the HCC-like IHC subtype is associated with HCC-related etiologic factors."

We added a new reference for this topic in ref. 30;

Jeon, Y.; Kwon, S.M.; Rhee, H.; Yoo, J.E.; Chung, T.; Woo, H.G.; Park, Y.N. Molecular and radiopathologic spectrum between HCC and intrahepatic cholangiocarcinoma. Hepatology. 2023, 77, 92-108.

[Inquiries 3]
3. Figure 5, the schematic diagram did not explain the down-stream function of non-inflamed phenotype.

[Response 3]

Thank you for the informative suggestion.

We added an additional statement in the legend of figure 5, page10, line 370-371, as follow;

" Consequently, lack of antigen-presenting machinery for the presentation of tumor-specific neoantigen result in the failure of attracting cytotoxic T lymphocytes (CTLs) into tumor."

[Inquiries 4]
4. If author can provide a detailed study on recent and previous clinical trials studies about CCA

[Response 4]

Thank you for the valuable suggestion.

We introduced the additional clinical trials in page 8-9, line 269-285 as follow;

" So far, several clinical trials have been conducted for the treatment of CCA.[2] Importantly, clinical trials studies reveal that two kinds of MTA, the FGFR tyrosine kinase inhibitor and the IDH1 inhibitor, were reported to be effective in treating advanced chemotherapy-resistant CCAs after progression on first-line systemic therapy with FGFR2 fusions/rearrangements and IDH1 mutations, respectively.[7-10]  In addition, HER2 and BRAF inhibitors are undergoing incorporation into the conventional systemic chemotherapy.[20]

Recently, treatment with ICI (durvalumab) in combination with gemcitabine and cisplatin showed superior efficacy to conventional first-line treatment with gemcitabine and cisplatin in overall survival and progression-free survival in patients with advanced CCAs.[11] In addition, two kinds of MTA, the FGFR tyrosine kinase inhibitor and the IDH1 inhibitor, were reported to be effective in treating advanced chemotherapy-resistant CCAs with FGFR2 fusions/rearrangements and IDH1 mutations, respectively.[7-10] The efficacy of ICIs should be affected by the TME, and the efficacy of MTAs depends on the mutation status of the target genes, understanding the TME and its association with driver mutations is necessary for chemotherapy management.[21-23]"

We added a new reference for this topic in ref. 20;

Uson Junior, P.L.S.; Bearss, J.; Babiker, H.M.; Borad, M.J. Novel precision therapies for cholangiocarcinoma: an overview of clinical trials. Expert Opin Investig Drugs. 2023, 32, 69-75.

Reviewer 2 Report

A small correction of the patients' description in the Table S1 (in Materials and Methods) is required: 29 patients had IHC, 1 listed as cholangiocarcinoma, and 2 listed as extrahepatic cholangiocarcinoma.

Author Response

To Reviewer 2

[Inquiries]

A small correction of the patients' description in the Table S1 (in Materials and Methods) is required: 29 patients had IHC, 1 listed as cholangiocarcinoma, and 2 listed as extrahepatic cholangiocarcinoma.

[Response]

Thank you for pointing put the error for the number of the cases.

We have corrected the statement in page3, line 105-107 as follow;

In this cohort, 29 patients had IHC and four had PHC. Among the remaining three, two were shown as extrahepatic cholangiocarcinoma and one was listed as cholangiocarcinoma, respectively.

Reviewer 3 Report

Based on TCGA transcriptome data, 36 CCAs were classified into inflamed (n=6) and non-inflamed (n=30) tumors with clustering markers including T cell surface markers, T cell signaling and activation-related markers, lymphocyte chemotactic markers, T cell-related cytolytic factors, and immune checkpoint molecules. The non-inflamed tumors were further grouped into non-inflamed A (n=17) and non-inflamed B (n=13). This study explored the association of genetic and epigenetic alterations with inflamed and non-inflamed CCAs, identifying that the methylation and gene expression levels of antigen-presenting machineries were critical for determining inflamed or non-inflamed phenotypes. The non-inflamed tumors had hypermethylation and downregulation of genes associated with antigen-presenting machineries. It is interesting that FGFR2 and IDH1/2 mutations had a “non-inflamed” phenotype. This study tried to explain how FGFR2 and IDH1/2 alterations impact on TME phenotypes by modulating the methylation antigen-presenting machineries. Overall, this manuscript provides some interesting findings.

Comments:

1.      The non-inflamed tumors were further divided into non-inflamed A and non-inflamed B. How to link the information of methylation and gene expression levels of antigen-presenting machineries to the subtypes?

2.      This manuscript was titled as “Non-inflamed tumor microenvironment and methylation/downregulation of antigen-presenting machineries in cholangiocarcinoma”, however the abstract described “In this study, we aimed to understand the molecular characteristics associated with inflamed phenotype of CCAs”. I would suggest to improve the abstract.

Author Response

To Reviewer 3

[Reviewer's overall comment]

This study tried to explain how FGFR2 and IDH1/2 alterations impact on TME phenotypes by modulating the methylation antigen-presenting machineries. Overall, this manuscript provides some interesting findings.

[Response]

Thank you for reviewing out manuscript carefully.

We have learned reviewer's comments and revised the manuscript as listed below.

[Inquiries 1]

  1. The non-inflamed tumors were further divided into non-inflamed A and non-inflamed B. How to link the information of methylation and gene expression levels of antigen-presenting machineries to the subtypes?

[Response 1]

Thank you for this comment for the comparison between non-inflamed A and non-inflamed B.

Through the multiple non-parametric comparisons among non-inflamed A, non-inflamed B, and inflamed group, we found the inflamed group shows the highest mRNA expression and the lowest DNA methylation level (Figure S3). In addition, median mRNA expression levels are lower in tumors in non-inflamed A than those in non-inflamed B for all 6 antigen-presenting machineries but TAP1 (Figure S3a). Similarly, median DNA methylation levels are higher in tumors in non-inflamed A than those in non-inflamed B in all 6 antigen-presenting machineries but B2M and CIITA (Figure S3b). Therefore, subgrouping of non-inflamed A and non-inflamed B is also associated the impaired expression of antigen-presenting machineries with DNA methylation.

For this presentation, we added an additional supplementary figure as Figure S3a and S3b.

We added the statement regarding this issue in result and discussion, as shown below;

Result, page 6, line 225-234

" Because we further divided the non-inflamed tumors into non-inflamed A and non-inflamed B, in which mRNA expressions of TIL-related markers were lower in non-inflamed A group than in non-inflamed B, we compared mRNA expressions and DNA methylation levels among the three groups. The inflamed group shows the highest mRNA expression and the lowest DNA methylation level (Figure S3). In addition, median mRNA expression levels are lower in tumors in non-inflamed A than those in non-inflamed B for all 6 antigen-presenting machineries but TAP1 (Figure S3a). Similarly, median DNA methylation levels are higher in tumors in non-inflamed A than those in non-inflamed B in all 6 antigen-presenting machineries but B2M and CIITA (Figure S3b)."

Discussion, page 10, line 340-343

" In addition, the majority of antigen-presenting machineries showed the lowest mRNA expressions and the highest DNA methylation levels in the tumors in the non-inflamed A group where downregulation of TIL-related markers was the most prominent (Figure S1 and S3)."

[Inquiries 2]

  1. This manuscript was titled as “Non-inflamed tumor microenvironment and methylation/downregulation of antigen-presenting machineries in cholangiocarcinoma”, however the abstract described “In this study, we aimed to understand the molecular characteristics associated with inflamed phenotype of CCAs”. I would suggest to improve the abstract.

[Response 2]

Thank you for the important comment.

We have corrected the start of the abstract as follow;

"Cholangiocarcinoma (CCA) is a refractory cancer; a majority of CCAs represents a non-inflamed tumor phenotype that should be resistant to the treatment including immune checkpoint inhibitors (ICIs). In this study, we aimed to understand the molecular characteristics associated with non-inflamed CCAs."
